# The Prevalence of Metabolic Syndrome and Undiagnosed Diabetes in Periodontitis Patients and Non-Periodontitis Controls in a Dental School

**DOI:** 10.3390/jcm13247512

**Published:** 2024-12-10

**Authors:** Madeline X. F. Kosho, Alexander R. E. Verhelst, Wijnand J. Teeuw, Sebastiaan van Bruchem, Kamran Nazmi, Victor E. A. Gerdes, Bruno G. Loos

**Affiliations:** 1Department of Periodontology, Academic Centre for Dentistry Amsterdam (ACTA), University of Amsterdam and Vrije Universiteit Amsterdam, 1081 LA Amsterdam, The Netherlands; alexanderverhelst@hotmail.com (A.R.E.V.); w.teeuw@vitalistopclinics.nl (W.J.T.); s.r.vanbruchem@uva.nl (S.v.B.); b.loos@acta.nl (B.G.L.); 2Department of Oral Biochemistry, Academic Centre for Dentistry Amsterdam (ACTA), University of Amsterdam and Vrije Universiteit Amsterdam, 1081 LA Amsterdam, The Netherlands; k.nazmi@acta.nl; 3Department of Vascular Medicine, Amsterdam University Medical Centre (AUMC), University of Amsterdam, 1105 AZ Amsterdam, The Netherlands; v.e.gerdes@amsterdamumc.nl; 4Department of Internal Medicine, Spaarne Gasthuis, 2134 TM Hoofddorp, The Netherlands

**Keywords:** metabolic syndrome, diabetes, HbA1c, periodontitis, obesity, fructosamine

## Abstract

**Background/Objectives**: Metabolic syndrome (MetS) and type 2 diabetes mellitus (T2DM) are major global health concerns, and they often go undetected. Periodontitis shares risk factors and is associated with both conditions. Assessing MetS risk factors among dental patients, especially those with periodontitis, may contribute to early detection and prompt treatment. However, current information about MetS prevalence rates in dental settings is limited. Therefore, our aim was to investigate the prevalence of MetS among patients with generalized periodontitis stage III/IV (GenPD), localized periodontitis stage III/IV (LocPD), and non-periodontitis controls. We also investigated the prevalence of undiagnosed T2DM in the same population. Moreover, we performed a pilot study to evaluate the possibility of measuring fructosamine levels in oral rinse samples, as an alternative to HbA1c, to screen for hyperglycemia. **Methods**: Periodontitis patients and non-periodontitis controls were recruited from a dental school, all aged ≥ 40 years. MetS prevalence was determined according to four different MetS definitions. T2DM prevalence was based on elevated HbA1c levels (HbA1c ≥ 7.0%). Biochemical analysis from blood was carried out by finger stick sampling. A subset of participants was asked to provide an oral rinse sample for the measurement of fructosamine, which was correlated to HbA1c from blood. **Results**: A total of 105 patients with periodontitis (GenPD stage III/IV: n = 44, LocPD stage III/IV: n = 61) and 88 non-periodontitis controls, with a mean age of 54.4 years, were included. The prevalence of MetS, according to four different MetS definitions, was 68.2–81.8% in GenPD patients, 42.6–62.3% in LocPD patients, and 52.3–69.3% in controls. The prevalence of T2DM, as evidenced by elevated HbA1c, was 20.5% in GenPD patients, 18.3% in LocPD patients, and 10.2% in controls (*p* = 0.094). A substantial number of subjects were not aware that they were being suspected to have T2DM, i.e., 13.6% in GenPD patients, 8.2% in LocPD patients, and 8.0% in controls (*p* = 0.335). In a subset of participants (n = 48), we found no significant correlation between HbA1c and oral fructosamine (r= 0.24, *p* = 0.103). **Conclusions**: The overall results showed a relatively high prevalence of patients with MetS and/or elevated HbA1c among periodontitis patients and controls in the dental school. Of those with suspected T2DM, a substantial number were not aware of having T2DM. Oral health care professionals could therefore contribute to early detection for T2DM and/or create awareness in patients at risk for a disease related to MetS. To date, initial results on the use of oral fructosamine as an alternative for HbA1c cast doubt, and further research is needed.

## 1. Introduction

Metabolic syndrome (MetS) is a major concern in global health. It is defined by a cluster of risk factors, consisting of abdominal obesity, increased blood pressure (BP), elevated blood levels of glucose, triglycerides (TG), and low high-density lipoprotein cholesterol (HDL-C) [1]. Individuals with MetS have a higher risk for type 2 diabetes mellitus (T2DM), cardiovascular disease (CVD), and premature death [2,3,4,5]. When CVD or T2DM is not present, MetS serves as a predictor for the development of these conditions [4]. Once CVD or T2DM occurs, MetS is often present, with the number of its components contributing to both disease progression and increased risk [3]. The global prevalence of MetS has been reported to be between 12.5 and 31.4% across different definitions for MetS [4]. In addition, the prevalence may be underestimated at present, as obesity and diabetes are also on the rise, particularly in developing countries [6,7,8].

The complexity of MetS and the clustering of its risk factors has led to several definitions of MetS with different diagnostic criteria, such as those from the World Health Organization (WHO) in 1998 [9], the National Cholesterol Education Program Adult Treatment Panel III (NCEP-ATP III) in 2001, which was revised in 2004 [10], and the International Diabetes Federation (IDF) in 2005 [11]. An attempt to unify the criteria of different organizations has led to a Joint Interim Statement (JIS) definition of MetS in 2009, which was issued by the IDF Task Force on Epidemiology and Prevention; the National Heart, Lung, and Blood Institute; the American Heart Association; the World Heart Federation; the International Atherosclerosis Society; and the International Association for the Study of Obesity [12].

Periodontitis is a chronic multifactorial inflammatory disease of the supporting structures of the teeth [13,14]. The worldwide prevalence of periodontitis is estimated to be 62%, while its severe form affects 11.2–23.6% of the worldwide population [15,16]. Several studies have shown that MetS and periodontitis are associated with an odds ratio (OR) ranging from 1.38 to 1.99 [17,18]. Activation of the same inflammatory pathways have been found in periodontitis, MetS, and obesity [19,20]. MetS may also be considered as a reflection of how the body responds to adipokines and cytokines secreted by visceral adipose tissue. These factors contribute to a systemic inflammatory burden which subsequently can affect insulin sensitivity and inflammation, as well as having direct metabolic effects [21].

Defining the prevalence of non-communicable disease risk, such as that for T2DM or CVD, in a dental setting gives insight into the overall health of such a population and provides opportunities for oral health care professionals to contribute to systemic disease detection and/or even prevention. A previous study in periodontitis patients showed that 18.1% of those with the severe form had suspected T2DM, and the corresponding figures for patients with mild/moderate periodontitis and for controls were 9.9% and 8.5%, respectively [22]. Furthermore, in another dental school study population, the prevalence of subjects with a very high risk for CVD mortality was 29.5% in patients with generalized periodontitis, 16.4% in patients with localized periodontitis, and 9.1% in controls [23]. These studies demonstrate that dental offices could be suitable locations to check patients for underlying and undetected diseases, particularly among periodontitis patients. Additionally, patients that are already aware of having T2DM or CVD, but still show abnormal blood values above their treatment goal (i.e., HbA1c, LDL cholesterol), can be reminded by their oral health professional to re-visit their doctor [23].

Currently, there is scarce information about the prevalence of MetS in patients that come to dental offices. Most studies have typically focused on the prevalence of periodontitis in populations with and without MetS [24,25,26,27] or focused on cross-sectional survey research to investigate the association between periodontitis and MetS [28,29]. One study, among dental patients from five Colombian dental university clinics, showed a MetS prevalence of 6.3% in periodontitis patients (n = 431) and 3.2% in controls (n = 220) [30]. Another study, among dental patients of the dental clinic of the Hebrew University-Hadassah School, showed a MetS prevalence of 47.7% in the periodontitis group (n = 385) and 6.0% in the control group (n = 119) [31]. However, these studies only used one specific definition for MetS, and periodontitis was not classified according to the World Workshop 2017 classification criteria [14]. Since a relatively high proportion of patients with T2DM and an increased CVD risk was found among periodontitis patients and controls in two dental school populations [22,23], we expect to observe a substantial proportion of patients with MetS. Furthermore, it is our hypothesis that patients with periodontitis may be more prone to having higher values of the clinical cardiometabolic parameters for MetS because of the large inflammatory burden in periodontitis and also because of the many shared risk factors between periodontitis, T2DM, and CVD [32,33,34].

Thus, the aim of our study was to evaluate the prevalence of MetS in the dental setting in patients with generalized periodontitis stage III/IV (GenPD), localized periodontitis stage III/IV (LocPD), and controls. We also aimed to investigate the prevalence of patients with elevated HbA1c levels, and identify those that may have undiagnosed T2DM. Finally, in a pilot experiment targeting in a subset of study participants, we aimed to investigate whether there is a potential for the use of oral rinse samples for the determination of signs of hyperglycemia.

## 2. Materials and Methods

### 2.1. Study Design and Recruitment

In this study, we included periodontitis patients who were referred to the Department of Periodontology at the Academic Centre for Dentistry of Amsterdam (ACTA) for diagnosis and treatment. Participants without periodontitis who visited the ACTA clinic for general dental care, such as check-ups or restorative procedures, were consecutively recruited as controls. All study subjects were enrolled between March 2018 and March 2020, with a minimum age of 40 years. At the time of study design, this minimum age was recommended to apply the intended screening instruments for cardiometabolic diseases (the European Systemic Coronary Risk Evaluation (SCORE) for CVD risk [35] and United States Preventive Services Taskforce (USPSTF) for T2DM [36]). This study population has been described before, and previously, we reported results on cardiovascular risk assessment [23]. Periodontitis patients from the former study had all stage III/IV severity and were classified as grade B or C. Originally, the study was designed as a cross-sectional study investigating the oral and systemic condition in periodontitis patients and controls (ClinicalTrials.gov Identifier NCT03459638, approved by the Medical Ethical Committee of the Amsterdam University Medical Center (2017.490 (A2019.151)-NL62337.029.17)). All participants received verbal and written information about the purpose of the study and gave their consent. The findings were reported in accordance with the STROBE guidelines [37].

During the first visit, periodontitis patients underwent a full-mouth periodontal examination performed by periodontists or residents from the Department of Periodontology. The following measurements were carried out for six sites per tooth using a manual periodontal probe (Williams probe, PW6, HuFriedy, Chicago, IL, USA): probing pocket depth (PPD, distance from the gingival margin to the bottom of the pocket), gingival recessions (exposure of root surfaces due to apical migration of the gingival margin), and clinical attachment loss (CAL, loss of the connective tissue attachment between the tooth and the supporting structures). Furthermore, tooth mobility (movement of the tooth horizontally and/or vertically) and furcation involvement (when supporting bone is lost around the branching point of a tooth with multiple roots) were measured. Dental radiographs (not older than ≤1 year old) were used to analyze interproximal alveolar bone levels.

Patients were initially screened for periodontitis using the Centers for Disease Control and Prevention–American Academy of Periodontology (CDC-AAP) case definition criteria [38]. Those who received a positive diagnosis—defined as having ≥2 interproximal sites with CAL ≥ 3 mm and ≥2 interproximal sites with PPD ≥ 4 mm, not on the same tooth, or one site with PPD ≥ 5 mm—were invited to participate [38]. Subsequently, for each periodontitis case, we applied staging (I–IV), grading (A, B, C), and determination of the extent (localized or generalized) per stage [14].

Control subjects were included if they: (1) did not meet the criteria for the case definition of periodontitis, (2) had no prior history of periodontal treatment, and (3) showed no interproximal alveolar bone loss on recent bitewing radiographs (≤1 year old). A distance of ≤3 mm between the cemento-enamel junction and the most coronal part of the radiographic alveolar crest was considered acceptable for inclusion for a non-periodontitis control.

### 2.2. Definitions of Metabolic Syndrome

In the literature, various definitions for MetS are available. This prompted us to analyze the data according to four published definitions for MetS (Table 1). Note, in the current study, we replaced fasting plasma glucose (FPG) with HbA1c, as has been used before [39]. A HbA1c measurement of ≥5.7% (39 mmol/mol) was used as the cut-off point for a hyperglycemic state, as is recommended by the American Diabetes Association (ADA) [40].

The following four definitions for MetS have been used (the corresponding MetS parameters are presented in Table 1):National Cholesterol Education Program Adult Treatment Panel III (NCEP-ATP III), which was revised in 2004 [10].National Cholesterol Education Program Adult Treatment Panel III (NCEP-ATP III) updated (2017), with age-adjusted blood pressure thresholds [41,42].International Diabetes Federation (IDF) (2005) [11].Joint Interim Statement (JIS) (2009) [12].

### 2.3. Definition for Suspected Diabetes

In addition to our aim to study the prevalence of MetS, we aimed to investigate the prevalence of patients with elevated HbA1c levels, and identify those that may have undiagnosed diabetes. An HbA1c threshold of ≥7.0% (53 mmol/mol) was used as the cut-off for the suspected presence of T2DM, a value that has shown 100% specificity in the Dutch population, thereby excluding possible false positive measurements [43].

### 2.4. Clinical Procedures

Self-reported questionnaires were used to collect data on age, sex, height, education level (primary, secondary, or >secondary, serving as proxy for socio-economic position), smoking habits, presence of CVD or a CVD event in the past, presence of T2DM, medication use, and physical activity.

A clinical examination was performed to assess blood pressure (BP) and waist circumference (WC). After the patient had been seated in the dental chair for at least 5 min. BP was measured three times on the right arm using a digital BP monitor (Omron^®^, Hoofddorp, The Netherlands). The average of the second and third readings was used for calculations. Body weight, measured with a digital scale, and self-reported height were used to calculate the body mass index (BMI). WC was measured at the midpoint between the lower edge of the last palpable rib and the top of the iliac crest, following a complete exhalation [44].

### 2.5. Blood Sample Collection and Analysis of Biochemical Values

Approximately 5–6 large drops of capillary blood were collected via finger stick into a microtube, which contained 17 USP/mL lithium heparin. This method was developed by Labonovum B.V., a certified chemical laboratory based in Rotterdam, The Netherlands [45]. The microtube was then sent to the laboratory by mail. Blood test results were made available on a secure digital platform and all patients received their blood test results. The biochemical markers analyzed in this study included HDL-cholesterol, triglycerides, and HbA1c.

### 2.6. Oral Fructosamine as an Alternative for Blood-Derived HbA1c

For a pilot in a subset of participants, oral rinse samples were collected from the last consecutively included 54 subjects of our study cohort to assess the level of oral fructosamine and to correlate them to blood values of HbA1c. The samples were collected in the same visit as the blood sample. Participants were not allowed to eat or brush their teeth for at least one hour prior to oral rinse collection. They were instructed to swallow once before the start of rinsing with sterile Dulbecco’s phosphate buffered saline (PBS; ThermoFisher Scientific, Waltham, MA, USA). The 10 mL PBS was supplied to the participant in a 50 mL centrifugation tube. All subjects were asked to rinse vigorously through all sides (left, right, upper, and lower) of the oral cavity for 30 s. After rinsing, they expectorated into a 30 mL medicine cup, and subsequently, the oral rinse was transferred back into the 50 mL centrifugation tube by the investigator. Immediately after collection, the oral rinse samples were cooled on ice. Subsequently, the oral rinse samples were vortexed for 10 s, divided into aliquots of 1.0 mL in 1.5 mL screw cap microtubes and stored at −80 °C. After thawing at room temperature, one aliquoted oral rinse sample per individual was centrifuged at 4 °C for 10 min. Fructosamine levels were measured with Enzyme-Linked Immunosorbent Assay kits (ELISA; Assaygenie; Dublin, Ireland) according to the instructions of the manufacturer.

### 2.7. Statistical Analysis

For the primary outcome (prevalence of MetS), no formal power calculation is available and the current results should be considered preliminary and explorative.

Data were analyzed with SPSS 26.0.0.0 (IBM SPSS, Chicago, IL, USA). Means, standard deviations, range, and frequency distributions were calculated. Missing data for some biochemical parameters were reported (indicated in the footnote of Table 2). Demographic, clinical, and dental characteristics, the parameters and the prevalence for MetS, were compared with an independent samples *t*-test or by chi-squared tests. ANOVA or chi-squared tests (two by two or linear by linear, where appropriate) were used when comparing three groups (non-periodontitis controls, LocPD patients, and GenPD patients). A Spearman’s correlation analysis was performed to assess the correlation between the HbA1c value and the oral rinse fructosamine concentration. Statistical significance was set at *p* < 0.05.

## 3. Results

### 3.1. Demographic, Clinical and Dental Characteristics of the Study Population

In total, 88 control subjects (mean age 54.8 years; 43.2% female), 61 patients with LocPD (mean age 54.2 years; 54.1% female), and 44 patients with GenPD (mean age 53.6 years; 34.1% female) were included in this study (Table 2). Less participants with GenPD (40.9%) had education beyond secondary level compared to LocPD (44.3%) and controls (60.2%) (*p* = 0.038). Smoking was highly prevalent among patients with GenPD (56.8%); patients with LocPD and controls smoked significantly less (24.6% and 10.2%, respectively; *p* < 0.001). The measurements for SystBP, DiastBP, BMI and BMI categories, and WC are presented in Table 2. It appears that the WC was the highest for GenPD (98.5 cm), and the lowest for controls (92.0 cm), with LocPD in between (96.3 cm) (*p* = 0.025). This trend was found to be reversed for the amount of days/week with exercise (*p* = 0.027). Subjects with GenPD had a lower number of present teeth (24.4) than the individuals with LocPD (26.6) and controls (27.4) (*p* < 0.001). All periodontitis patients had stage III/IV severity and were either classified as grade B (32.4%) or grade C (67.7%).

### 3.2. MetS Parameters According to the Four Definitions for MetS

Table 3 shows the prevalence of subjects that scored positive for MetS parameters according to the various definitions for MetS. When the cut-off for WC was lowered (male ≥ 94 cm, female ≥ 80 cm), the prevalence for an increased WC was significantly different between controls (63.6%), LocPD (78.7%) and GenPD (81.8%) (*p* = 0.016).

### 3.3. Prevalence of Metabolic Syndrome

In the current study, we applied four definitions (Table 1) to assess the prevalence of MetS in patients with LocPD stage III/IV, patients with GenPD stage III/IV, and non-periodontitis controls, recruited in a dental school. These results are presented in Table 4. In general, irrespective of the investigated definitions for MetS, the prevalence for MetS ranged from 42.6% to 81.8%. The lowest prevalence of MetS was seen in LocPD patients (42.6%), based on the revised and updated definition of NCEP-ETP III. Otherwise, irrespective of each definition, more than half of all study participants apparently suffered from MetS. The highest prevalence (81.8%) of MetS was observed for GenPD patients, applying the definition according to JIS. For all MetS definitions, the patients with GenPD showed the highest prevalence, ranging from 68.2% to 81.8%. Using the definition of IDF, it was shown that the lowest prevalence of MetS was found in non-periodontitis controls (55.7%) and the highest prevalence was found in GenPD patients (70.5%), while the patients with LocPD patients showed a prevalence in between (60.7%) (*p* = 0.109). In contrast, applying the Revised NCEP-ATP III, the patients with GenPD showed the highest prevalence (79.5%) at a statistically significant level compared to non-periodontitis controls (58.0%) and LocPD patients (55.7%) (*p* = 0.034). Within the comparison between control subjects vs. GenPD patients, the MetS prevalence was significantly higher in the GenPD patients compared to controls when applying the Revised NCEP-ATP III definition (*p* = 0.014). When the control and LocPD group were combined and compared with the GenPD group, we noted that MetS was significantly higher in the GenPD group when applying the MetS definition according to Revised NCEP-ATP III and Revised NCEP-ATP III updated (Table 4).

### 3.4. Prevalence of Subjects with Elevated HbA1c ≥ 7.0% (≥53 mmol/mol)

Figure 1 shows the prevalence of subjects with elevated HbA1c ≥ 7.0% (≥53 mmol/mol). It depicts a statistically non-significant trend of 10.2% in controls, 18.3% in LocPD stage III/IV patients, and 20.5% in GenPD stage III/IV patients (*p* = 0.094). Interestingly, the prevalence of patients with suspected T2DM (HbA1c ≥ 7.0%) that were not aware of having T2DM was 8.0% in controls, 8.2% in LocPD, and 13.6% in GenPD (*p* = 0.153). There was no further significant difference, when comparisons were made between controls vs. LocPD + GenPD (*p* = 0.083), between controls vs. GenPD (*p* = 0.107), or between controls + LocPD vs. GenPD (*p* = 0.259).

### 3.5. Correlation Between HbA1c and Oral Fructosamine

We explored whether a non-invasive test, other than a blood finger stick, could be used in a dental office to screen for hyperglycemia in dental patients. To this end, we collected oral rinse samples from a subset of patients to assess fructosamine levels as an alternative to HbA1c. From 6 of the original 54 subjects with available oral rinse samples, the fructosamine levels were below detection level. Figure 2 presents a scatter plot of the HbA1c and fructosamine values for the remaining 48 subjects (12 controls, 17 LocPD stage III/IV, and 19 GenPD stage III/IV). There was no significant correlation between finger-stick derived HbA1c (%) and oral rinse levels of fructosamine (µmol/L) (r = 0.24, *p* = 0.103).

## 4. Discussion

The prevalence of MetS was high, ranging between 52.3 and 69.3% for non-periodontitis controls, 42.6 and 62.3% for LocPD patients, and 68.2 and 81.8% in GenPD patients according to different MetS definitions. GenPD patients showed a significantly higher MetS prevalence compared to controls and LocPD patients, according to the revised NCEP-ATP III classification.

Secondly, the study also showed a prevalence of suspected T2DM of 10.2% in non-periodontitis controls, 18.3% in LocPD patients, and 20.5% in GenPD patients. Among these patients, a substantial number were not aware of potentially having diabetes: 8.0% in non-periodontitis controls, 8.2% in LocPD patients, and 13.6% in GenPD patients.

Furthermore, there was no correlation observed between HbA1c (%) and oral fructosamine (µmol/L), indicating that oral rinse fructosamine measurements are, as yet, of doubtful use as a screening instrument in a dental setting. This corroborates the findings of a systematic review [46].

The current study provides new insights into the prevalence of MetS and T2DM among dental patients with LocPD, with GenPD, and those without periodontitis, utilizing the World Workshop 2017 classification criteria for periodontitis [14]. Over half of our study participants aged ≥ 40 years, and an even higher proportion among those with periodontitis, were affected by MetS. The MetS prevalence in all our study groups (control, LocPD, and GenPD) was higher compared to the Dutch prevalence of MetS of 29.2% in subjects aged 45–65 years (n = 6602) [47]. In that study, hypertension was the most prevalent MetS parameter, affecting 61.7%, which was comparable to the prevalence observed in our cohort, with rates of 55.7% in controls, 63.9% in the LocPD group, and 68.2% in the GenPD group [47]. Other MetS parameters in the Dutch cohort showed a prevalence of 40.0% for abdominal obesity (WC ≥ 102 cm for males or ≥88 cm for females), 30.9% for hyperglycemia (fasting glucose > 5.6 mmol/L or treatment), 25.3% for hypertriglyceridemia (TG ≥ 1.7 mmol/L or treatment), and 23.2% for low HDL-cholesterol (HDL-C < 1.00 mmol/L for males or <1.30 mmol/L for females) [47]. Our cohort showed a higher prevalence of abdominal obesity among periodontitis patients, while the prevalence in the control group was similar to the Dutch cohort. Further comparison with the Dutch cohort showed that both our periodontitis group and control group had approximately twice as many patients with hyperglycemia, low HDL-cholesterol, and hypertriglyceridemia [47].

Furthermore, in comparison with the Dutch T2DM prevalence rate of 5.0% among patients (40–59 years) that visit their general physician, the prevalence rates of T2DM in our study were higher for all study groups [48]. Notably, when measuring HbA1c, a substantial number of the subjects appeared to be unaware of being suspected to have T2DM, which was also shown in our previous study [22]. This corroborates the results of a recent global survey of the International Diabetes Federation that showed consistently that close to 50% of people living with T2DM are undiagnosed [7]. Additionally, the mean age at which T2DM is diagnosed in the Netherlands is 60.3 years, whereas in our study, the mean age of patients with an elevated HbA1c level who were unaware of T2DM was 56.7 years [48]. This indicates that patients might be screened earlier in a dental office than in a general practice. Moreover, the results of this study showed that approximately one fifth of the periodontitis patients had elevated HbA1c levels. The literature has shown strong associations between periodontitis and T2DM [33,49,50,51,52]. In fact, periodontitis is considered a complication of T2DM, alongside pathologies in the eyes, kidneys, and extremities, as well as neuropathies [53]. Furthermore, in previous studies in our dental school as well as in other settings in other countries, the prevalence of T2DM was consistently increased in periodontitis patients [22,29,54].

The above findings underscore the necessity for dental professionals to be aware that a significant number of their patients may have underlying systemic conditions, such as being overweight or at risk of developing T2DM. Such awareness should encourage dental professionals to consider the overall healthcare needs of their patients, aiming to participate in a holistic care approach towards both oral and general health. In addition to potential of screening for underlying systemic conditions, dental professionals can play a crucial role in reminding patients who are already aware of their condition to consult their doctor when abnormal blood values are detected.

We suggest that the current study population visiting the dental school for their dental care and periodontal treatment is, in general, less healthy than the average Dutch citizen, which could perhaps be related to a lower socio-economic position; there might be an overrepresentation of dental patients at the dental school from low socio-economic backgrounds, and those who partake in unfavorable lifestyle habits such as smoking [55]. In the GenPD patients group, the prevalence of smoking was 56.8%, which is considerably higher than the smoking prevalence in the general Dutch population of approximately 20% [56]. Further, the prevalence of overweight and obesity in the study population was notably higher compared to the general Dutch population, where 35% are classified as overweight and 16% as obese [56]. Moreover, globally, the prevalence of obesity and T2DM continues to increase [6,7,8]. Another study among 37,801 individuals, conducted in the same dental school, observed that the overall prevalence of self-reported comorbidity (≥1 systemic disease) was 35.6%, while the prevalence of self-reported multimorbidity (≥2 systemic diseases) was 18.4% [55]. Among individuals ≥ 55 years old with periodontitis, the frequency rates were notably higher, with comorbidity at 59.2% and multimorbidity at 33.9% [55]. Obviously, when objective measurements like in the current study are taken among dental school patients, the prevalence of morbidities will be more frequent, as we have found here.

A strength of this study is the use of four different definitions of MetS. Previous research has demonstrated that different definitions of MetS may result in substantially different MetS prevalences [57]. To be able to compare the MetS prevalence with other studies, it is necessary to use the same criteria to define MetS; however, a gold standard definition for MetS is still lacking. Another strength is that we also looked at the difference in prevalence among the extent of periodontitis (LocPD, GenPD). The prevalence of MetS in GenPD patients was higher compared to the control group (according to the revised NCEP-ATP III) and compared to the controls combined with LocPD patients (according to the revised NCEP-ATP III and its updated definition). We suggest that patients with a large inflammatory burden may be more prone to having higher values for clinical cardiometabolic risk factors and may suffer more often from MetS [32,58].

A limitation in the present study regarding the definition of MetS is the lack of fasting glucose measurements that are formally required to determine the presence or absence of MetS. Instead, we used HbA1c to indicate chronic hyperglycemia. This replacement of fasting glucose with HbA1c has been performed before and has been proven a valid alternative [39]. The available values for HbA1c served very well to assess the prevalence of suspected T2DM.

Furthermore, regarding our pilot findings about oral fructosamine as an alternative to HbA1c, it needs to be considered that only 48 subjects provided oral rinse samples. Unfortunately, and as expected in this mainly non-diabetic population, many of these samples were in a limited range of HbA1c. For future studies, more samples with higher HbA1c levels (>6.0%) are needed to increase the range. Such studies are in progress, but nevertheless, these initial results cast doubt on oral fructosamine as an alternative to Hba1c. Other studies have reported promising results showing correlations between fructosamine and HbA1c [59,60]. One study found a good correlation between oral fructosamine and HbA1c among diabetes patients, but moderate correlations among other patients [59].

## 5. Conclusions

The overall results show a high prevalence of MetS among periodontitis patients and controls in our dental school. Dental professionals should be aware of the criteria of MetS and the prevalence of MetS in their settings. Knowing that someone suffers from MetS or has risk factors for MetS is important, because it increases their risk of developing serious conditions such as CVD, T2DM, and cerebrovascular accidents [51]. Dental professionals can participate in a holistic approach into the overall health of their patients and bring lifestyle changes to attention of their patients, which will improve not only their oral health, but their overall health. They may have a great impact on contributing to the general health of their patients. An example of how dental professionals can contribute to general health interventions in the dental setting is provided by Making Every Contact Count (MECC), a project from Public Health England and NHS England [33,61]. Furthermore, dental professionals can actively participate in screening for undiagnosed T2DM. Currently, HbA1c assessments via a finger stick is the standard. Although we anticipate the future use of oral rinse samples as a non-invasive alternative, it is uncertain whether fructosamine is able to replace Hba1c measurement, and therefore further research is needed.

## Figures and Tables

**Figure 1 jcm-13-07512-f001:**
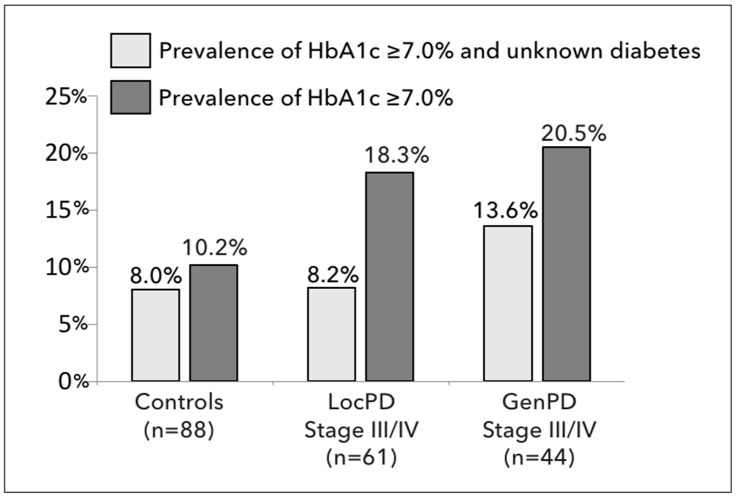
Prevalence of patients with HbA1c levels ≥ 7.0% (≥53 mmol/mol) in controls, patients with localized periodontitis (LocPD) stage III/IV, and patients with generalized periodontitis (GenPD) stage III/IV. Dark grey columns depict the prevalence of patients with HbA1c ≥ 7.0% (overall *p*-value = 0.094). Light grey columns depict the prevalence of patients that had suspected T2DM (HbA1c ≥ 7.0%), but were unaware of having T2DM (overall *p*-value = 0.335).

**Figure 2 jcm-13-07512-f002:**
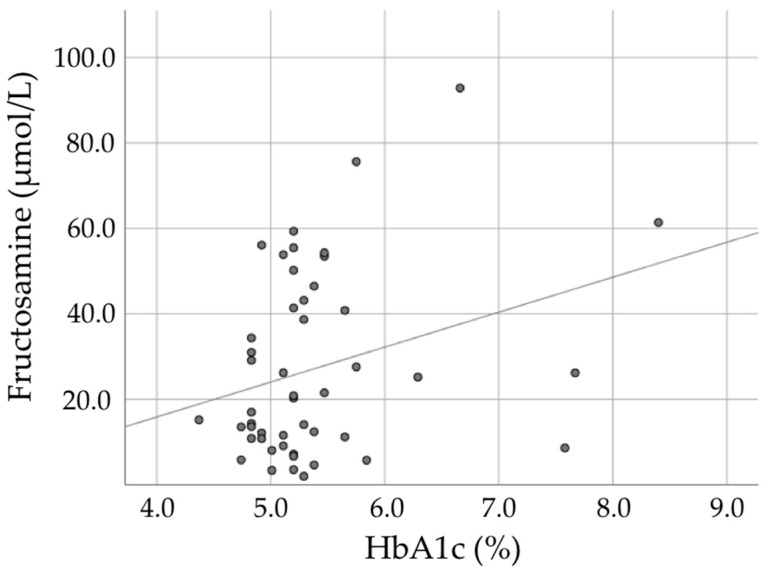
Correlation between HbA1c (%) and oral fructosamine (µmol/L) among a subset of the study population (n = 48) (r = 0.24, *p* = 0.103).

**Table 1 jcm-13-07512-t001:** Criteria for four different definitions of metabolic syndrome.

MetS Parameters	Revised NCEP-ATP III (2004)	Revised NCEP-ATP III Updated (2017)	IDF (2005)	JIS (2009)
Waist circumference (WC)	M ≥ 102 cm F ≥ 88 cm	M ≥ 102 cm F ≥ 88 cm	M ≥ 94 cm F ≥ 80 cm	M ≥ 94 cm F ≥ 80 cm
Blood pressure (BP) (mmHg)	SystBP ≥ 130 and/or DiastBP ≥ 85 and/or treatment	<60 years: ≥140/90 ≥60 years ≥150/90 and/or treatment	SystBP ≥ 130 and/or DiastBP ≥ 85 and/or treatment	SystBP ≥ 130 and/or DiastBP ≥ 85 and/or treatment
HbA1c	≥5.7% (39 mmol/mol) and/or treatment and/or diagnosed T2DM	≥5.7% (39 mmol/mol) and/or treatment and/or diagnosed T2DM	≥5.7% (39 mmol/mol) and/or treatment and/or diagnosed T2DM	≥5.7% (39 mmol/mol) and/or treatment and/or diagnosed T2DM
Triglycerides (TG)	≥1.7 mmol/L and/or treatment	≥1.7 mmol/L and/or treatment	≥1.7 mmol/L and/or treatment	≥1.7 mmol/L and/or treatment
HDL-C	M < 1.03 mmol/L F < 1.30 mmol/L and/or treatment	M < 1.03 mmol/L F < 1.30 mmol/L and/or treatment	M < 1.03 mmol/L F < 1.29 mmol/L and/or treatment	M < 1.0 mmol/L F < 1.3 mmol/L and/or treatment

Criteria to define MetS	≥3 MetS parameters	≥3 MetS parameters	WC above cut-off value and ≥3 MetS parameters	≥3 MetS parameters

WC: waist circumference; M: male; F: female; BP: blood pressure; SystBP: systolic blood pressure; DiastBP: diastolic blood pressure; HbA1c: glycated haemoglobin; T2DM: type 2 diabetes mellitus; HDL-C: high-density lipoprotein cholesterol; TG: triglycerides; NCEP-ATP III: National Cholesterol Education Program Adult Treatment Panel III; IDF: International Diabetes Federation; JIS: Joint Interim Statement.

**Table 2 jcm-13-07512-t002:** Demographic, clinical, and dental characteristics of the study population.

	Control (n = 88)	Periodontitis Localized Stage III/IV (n = 61)	Periodontitis Generalized Stage III/IV (n = 44)	*p*-Value
Age (years)	54.8 ± 9.6	54.2 ± 8.9	53.6 ± 8.6	0.776
Sex				0.537
Male	50 (56.8)	28 (45.9)	29 (65.9)	
Female	38 (43.2)	33 (54.1)	15 (34.1)	
Education ^†^				0.038
Primary	11 (12.5)	15 (24.6)	8 (18.2)	
Secondary	24 (27.3)	19 (31.1)	18 (40.9)	
>Secondary	53 (60.2)	27 (44.3)	18 (40.9)	
Smoking status				<0.001
Current	9 (10.2)	15 (24.6)	25 (56.8)	
Former	29 (33.0)	24 (39.3)	12 (27.3)	
Never	50 (56.8)	22 (36.1)	7 (15.9)	
SystBP (mmHg)	130 ± 20.3	132 ± 8.0	134 ± 15.7	0.546
DiastBP (mmHg)	83 ± 11.9	83 ± 9.9	84 ± 9.5	0.845
BMI (kg/m^2^)	26.0 ± 4.6	26.9 ± 4.5	27.4 ± 4.1	0.186
BMI category (kg/m^2^)				
<25	41 (46.6)	26 (42.6)	13 (29.5)	0.061
≥25 and <30 ^‡^	34 (38.6)	19 (31.1)	21 (47.7)	
≥30 ^‡^	13 (14.8)	16 (26.2)	10 (22.7)	
Waist circumference (cm)	92.0 ± 15.1	96.3 ± 12.2	98.5 ± 12.9	0.025
Days/week with exercise (>30 min)	4.8 ± 2.4	3.9 ± 2.4	3.6 ± 2.8	0.027
HDL-C (mmol/L) ^‡‡^	1.10 ± 0.5	1.22 ± 0.5	1.18 ± 0.4	0.246
Triglycerides (mmol/L) ^‡‡^	1.81 ± 1.0	2.03 ± 1.2	2.06 ±1.4	0.404
Self-reported CVD	7 (8.0)	8 (13.1)	8 (18.1)	0.082
Self-reported DM	3 (3.4)	6 (9.8)	4 (9.1)	0.153
Mean HbA1c (%) (mmol/mol) ^¶^				
All subjects	6.1 ± 0.8	6.0 ± 0.8	6.2 ± 1.1	0.778
	(42.8 ± 8.8)	(42.6 ± 8.9)	(43.8 ± 11.8)	
# Teeth	27.4 ± 2.2	26.6 ± 2.3	24.4 ± 3.4	<0.001
# Teeth with ≥33% bone loss	NA	2.2 ± 1.5	6.6 ± 3.0	NA
# Teeth with PPD ≥ 6 mm	NA	5.5 ± 4.0	13.0 ± 6.4	NA
# Sites with PPD ≥ 6 mm	NA	10.4 ± 9.4	30.8 ± 22.4	NA

Note: data are presented as the mean ± SD or as n (%). ^†^ Primary: primary education or preparatory secondary vocational education; Secondary: higher secondary general education, pre-university education; >Secondary: beyond secondary education; CVD: cardiovascular disease; SystBP: systolic blood pressure; DiastBP: diastolic blood pressure; BMI: body mass index; ^‡^ BMI determined overweight (≥25 kg/m^2^) and obesity (≥30 kg/m^2^); HDL-C: high density lipoprotein cholesterol; HbA1c: glycated haemoglobin; #: number of; PPD: probing pocket depth; NA: not applicable. Missing values for: ^‡‡^ Control: n = 1; Periodontitis generalized stage III/IV: n = 1; ^¶^ Periodontitis localized stage III/IV: n = 1.

**Table 3 jcm-13-07512-t003:** MetS parameters according to the four definitions for MetS ^1^.

	Control (n = 88)	Periodontitis Localized Stage III/IV (n = 61)	Periodontitis Generalized Stage III/IV (n = 44)	*p*-Value
WC: M ≥ 102/F ≥ 88 cm	34 (38.6)	31 (50.8)	24 (54.5)	0.062
WC: M ≥ 94/F ≥ 80 cm	56 (63.6)	48 (78.7)	36 (81.8)	0.016
BP: ≥130/85 mmHg and/or treatment	49 (55.7)	39 (63.9)	30 (68.2)	0.143
BP < 60 years: ≥140/90 mmHg or BP ≥ 60 years: ≥150/90 mmHg and/or treatment	36 (40.9)	22 (36.1)	22 (50.0)	0.437
HbA1c ≥ 5.7% (39 mmol/mol) and/or treatment and/or diagnosed DM	59 (67.0)	36 (60.0)	28 (63.6)	0.596
HDL-C: M < 1.03/F < 1.30 mmol/L and/or treatment	56 (63.6)	33 (54.1)	28 (65.1)	0.914
HDL-C: M < 1.00/F < 1.30 mmol/L and/or treatment	56 (63.6)	32 (52.5)	26 (60.5)	0.544
TG ≥ 1.7 mmol/L and/or treatment	40 (45.5)	35 (57.4)	26 (60.5)	0.078
Mean amount of MetS parameters Revised NCEP-ATP III	2.7 ± 1.2	2.9 ± 1.3	3.1 ± 1.2	0.252
Mean amount of MetS parameters Revised NCEP-ATP III updated	2.6 ± 1.3	2.6 ± 1.4	2.9 ± 1.2	0.302
Mean amount of MetS parameters IDF	2.3 ± 1.0	2.4 ± 1.1	2.6 ± 1.1	0.443
Mean amount of MetS parameters JIS	3.0 ± 1.2	3.1 ± 1.2	3.3 ± 1.2	0.241

Note. Data are presented as the mean ± SD or as n (%). ^1^ See Table 1; abbreviations also see Table 1. WC: waist circumference; M: male; F: female; BP: blood pressure; HbA1c: glycated hemoglobin; HDL-C: high density lipoprotein cholesterol; TG: triglycerides; NCEP-ATP III: National Cholesterol Education Program Adult Treatment Panel III; IDF: International Diabetes Federation; JIS: Joint Interim Statement.

**Table 4 jcm-13-07512-t004:** Prevalence of metabolic syndrome.

	Control (n = 88)	Periodontitis Localized Stage III/IV (n = 61)	Periodontitis Generalized Stage III/IV (n = 44)	Overall *p*-Value Between 3 Groups	*p*-Value Between Control vs. GenPD	*p*-Value Between Control + LocPD vs. GenPD
Revised NCEP-ATP III	51 (58.0)	34 (55.7)	35 (79.5)	0.034	0.014	0.007
Revised NCEP-ATP III updated	46 (52.3)	26 (42.6)	30 (68.2)	0.189	0.081	0.020
IDF	49 (55.7)	37 (60.7)	31 (70.5)	0.109	0.102	0.129
JIS	61 (69.3)	38 (62.3)	36 (81.8)	0.255	0.125	0.051

Note. Data are presented as the mean ± SD or as n (%). NCEP-ATP III: National Cholesterol Education Program Adult Treatment Panel III; IDF: International Diabetes Federation; JIS: Joint Interim Statement.

## Data Availability

The raw data supporting the conclusions of this article will be made available by the authors on request.

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
