# Peer review of "The Prevalence of Metabolic Syndrome and Undiagnosed Diabetes in Periodontitis Patients and Non-Periodontitis Controls in a Dental School"

_jcm, 2024, doi:10.3390/jcm13247512_

Round 1
Reviewer 1 Report
Comments and Suggestions for Authors
The authors investigated the prevalence of metabolic syndrome and undiagnosed diabetes among patients of the dental school. Both of these were surprisingly high, and the authors conclude that the dentists could contribute to early detection of diabetes among their patients. However, the oral rinse to measure fructosamine was not proven as a suitable biomarker of elevated HbA1c levels.
Minor comments:
1. Methods: please, describe here only the methods, not hypothesis nor results.
2. MetS definition number 2. It would be logical that you present the criteria only in Table 1, not both the table and text.
3. The risk factors (Table 1) are often called as “positive MetS parameters”.
4. Remove the first paragraph of chapter 2.8. just leaving the statement that formal power calculations were not done.
5. Was the correlation between HbA1c and oral rinse really investigated using a linear regression analysis or correlation analysis (Pearson or Spearman)?
6. Remove the non-significant results from chapter 3.1. The reader will find them in Table 2.
7. The same applies for chapter 3.2.
8. Maybe the comparisons could be also done for controls + LocPD vs. GenPD.
9. Please, start the Discussion with the main findings.
Reviewer 2 Report
Comments and Suggestions for Authors
Specific Comment:
• Line 108: Could the authors clarify the rationale for setting the minimum age at 40 years?
General Comment:
This manuscript is thoroughly conducted and exceptionally well-written, especially in the introduction, where the authors provide essential background information and establish a clear context for the study. The methodology is comprehensive, adhering to STROBE guidelines, with correct identification of the study type and a detailed timeline of data collection. The authors have also ensured ethical standards through study registration and patient consent. The results section is clearly presented, making the study’s findings accessible and impactful. Given these strengths, I recommend acceptance of this manuscript upon clarification of the specific comment above.
Author Response
Please see the attachment
Dear Reviewer,
Thank you very much for the thorough review of the manuscript. The reviews and suggestions have helped substantially to improve the manuscript. We are pleased to attach the revised version of our manuscript, entitled “The prevalence of metabolic syndrome and undiagnosed diabetes in periodontitis patients and non-periodontitis controls in a dental school”. We appreciate that the manuscript is of interest, and based on the reviews, we understand the need for additional revisions and clarifications. Please find below a point-by-point response to the comments and suggestions. The changes in the manuscript have been highlighted in the text. We hope that we have properly addressed the comments and advice, and that the manuscript will now be suitable for publication in the “Journal of Clinical Medicine”.
Yours sincerely,
On behalf of all authors
Madeline Kosho
|
3. Point-by-point response to Comments and Suggestions for Authors
|
|
Comment 1: This manuscript is thoroughly conducted and exceptionally well-written, especially in the introduction, where the authors provide essential background information and establish a clear context for the study. The methodology is comprehensive, adhering to STROBE guidelines, with correct identification of the study type and a detailed timeline of data collection. The authors have also ensured ethical standards through study registration and patient consent. The results section is clearly presented, making the study’s findings accessible and impactful. Given these strengths, I recommend acceptance of this manuscript upon clarification of the specific comment above.
|
|
Response 1: Dear reviewer, we appreciate your view on our manuscript. Thank you for your kind words.
|
|
Comment 2: Line 108: Could the authors clarify the rationale for setting the minimum age at 40 years?
Response 2: Thank you for your comment. During the time of research proposal, the minimum screening age for diabetes was 45 years according to the ADA (American Diabetes Association), 45 years according to the Dutch General Practitioner Guidelines and 40 years according to the USPSTF (United States Preventive Services Taskforce). In our former study (Kosho et al. 2023) we have calculated the 10-year risk on a cardiovascular event according to the European SCORE assessment, where also a minimum age of 40 years was required. Taking in consideration the various guidelines, we had decided to set our minimum age at 40 years at time of the study design.
We amended 2.1. Study design and recruitment (page 3). It reads now as follows:
At the time of study design, this minimum age was recommended to apply the intended screening instruments for cardiometabolic diseases (the European Systemic Coronary Risk Evaluation (SCORE) for CVD risk [35] and United States Preventive Services Taskforce (USPSTF) for T2DM [36]). |

Reviewer 3 Report
Comments and Suggestions for Authors
Dear Authors,
Congratulations on the job you have done and presented in this manuscript. I believe that your work is significant to the field and can be o high interest for the general reader. However, for the moment there are some modifications required prior to consideration for publication of your paper in a such high quality journal.
1. First and foremost, there are some grammar and spelling errors within the text. please revise carefully.
2.Abstract - well written and structured, maybe the authors can add a short ( 1-2 lines) introduction before the aim.
3 Introduction- this section is comprehensive with good referencing, although some references are quite old ( more than 20 years). I would suggest the authors to update the references list. Furthermore, the aim of the study should be written at the end of the section.
4 Material and method- this is by far the strongest part of your paper. I like a lot how the authors structured the section. There are some minor details that can increase the visibility for the general reader in this part: what was the mean CAL? what was the mean of gingival recessions? furcation involvement? and according to what classification..... " manual probe"- what type of periodontal probe was used in the study group?
5. Results are clearly presented, no modifications are required
6. There are some large paragraphs that require some referencing to support the statements. The authors should also add the strengths of the present research. please also check the iThenticate report, since it's quite high.
&. Reference style is not consistent, please revise. Check also the format of the year of publication: most are written in bold, some are not.
Author Response
Please see the attachment
Dear Reviewer,
Thank you very much for the thorough review of the manuscript. The reviews and suggestions have helped substantially to improve the manuscript. We are pleased to attach the revised version of our manuscript, entitled “The prevalence of metabolic syndrome and undiagnosed diabetes in periodontitis patients and non-periodontitis controls in a dental school”. We appreciate that the manuscript is of interest, and based on the reviews, we understand the need for additional revisions and clarifications. Please find below a point-by-point response to the comments and suggestions. The changes in the manuscript have been highlighted in the text. We hope that we have properly addressed the comments and advice, and that the manuscript will now be suitable for publication in the “Journal of Clinical Medicine”.
Yours sincerely,
On behalf of all authors
Madeline Kosho
|
3. Point-by-point response to Comments and Suggestions for Authors
Congratulations on the job you have done and presented in this manuscript. I believe that your work is significant to the field and can be o high interest for the general reader. However, for the moment there are some modifications required prior to consideration for publication of your paper in a such high quality journal.
|
|
Comment 1: First and foremost, there are some grammar and spelling errors within the text. please revise carefully.
Response 1: Thank you for your remark. We have corrected our grammar and spelling errors throughout the text in our manuscript.
Comment 2: Abstract - well written and structured, maybe the authors can add a short ( 1-2 lines) introduction before the aim.
Response 2: Thank you for your thoughtful feedback. We have modified our abstract by adding a short introduction before the aim (page 1).
It now reads as follows:
Background/Objectives: Metabolic syndrome (MetS) and Type 2 Diabetes Mellitus (T2DM) are major global health concerns and they often go undetected. Both conditions share risk factors and are associated with periodontitis. Assessing MetS risk factors among dental attenders, especially those with periodontitis, may contribute to early detection and prompt treatment. However, current information about MetS prevalence rates in dental settings is limited. Therefore, our aim was to investigate the prevalence of MetS among patients with generalized periodontitis stage III/IV (GenPD), localized periodontitis stage III/IV (LocPD) and non-periodontitis controls in our dental school. Also, we investigated the prevalence of undiagnosed T2DM in the same population. Moreover, we performed a pilot study to evaluate the possibility of measuring fructosamine levels in oral rinse samples, as alternative for HbA1c, to screen for hyperglycemia.
Comment 3: Introduction- this section is comprehensive with good referencing, although some references are quite old ( more than 20 years). I would suggest the authors to update the references list. Furthermore, the aim of the study should be written at the end of the section
Response 3: Thank you for your remark. We have modified our references by replacement with more recent references, where possible. However the following references were not possible to update, as these references describe statements for the definitions for MetS.
9. Alberti, K.G.M.M.; Zimmet, P.Z.; WHO Consultation Definition, Diagnosis and Classification of Diabetes Mellitus and Its Complications. Part 1: Diagnosis and Classification of Diabetes Mellitus. Provisional Report of a WHO Consultation. Diabet. Med. 1998, 15, 539–553, doi:10.1002/(SICI)1096-9136(199807)15:7<539::AID-DIA668>3.0.CO;2-S. 10. Grundy, S.M.; Brewer, H.B.; Cleeman, J.I.; Smith, S.C.; Lenfant, C. Definition of Metabolic Syndrome: Report of the National Heart, Lung, and Blood Institute/American Heart Association Conference on Scientific Issues Related to Definition. Circulation 2004, 109, 433–438, doi:10.1161/01.CIR.0000111245.75752.C6. 11. Alberti, K.G.M.M.; Zimmet, P.; Shaw, J. Metabolic Syndrome—a New World‐wide Definition. A Consensus Statement from the International Diabetes Federation. Diabetic Medicine 2006, 23, 469–480, doi:10.1111/j.1464-5491.2006.01858.x. 12. Alberti, K.G.M.M.; Eckel, R.H.; Grundy, S.M.; Zimmet, P.Z.; Cleeman, J.I.; Donato, K.A.; Fruchart, J.-C.; James, W.P.T.; Loria, C.M.; Smith, S.C. Harmonizing the Metabolic Syndrome: A Joint Interim Statement of the International Diabetes Federation Task Force on Epidemiology and Prevention; National Heart, Lung, and Blood Institute; American Heart Association; World Heart Federation; International Atherosclerosis Society; and International Association for the Study of Obesity. Circulation 2009, 120, 1640–1645, doi:10.1161/circulationaha.109.192644.
Furthermore, we wanted to improve the introduction by elaborating more on the available studies about MetS prevalence among periodontitis patients and controls in dental offices (page 2, 3). It reads now as follows:
Currently, there is scarce information about the prevalence of MetS in patients that come to dental offices. Most studies have typically focused on the prevalence of periodontitis in populations with and without MetS [24–27] or focused on cross-sectional survey research to investigate the association between periodontitis and MetS [28,29]. One study, among dental attenders of five Colombian dental university clinics, showed a MetS prevalence of 6.3% in periodontitis patients (n=431) and 3.2% in controls (n=220) [30]. Another study, among dental attenders of the dental clinic of the Hebrew University-Hadassah School, showed a MetS prevalence of 47.7% in the periodontitis group (n=385) and 6.0% in the control group (n=119) [31]. However, these studies only used one specific definition for MetS and periodontitis was not classified according World Workshop 2017 classification criteria [14].
Comment 4: Material and method- this is by far the strongest part of your paper. I like a lot how the authors structured the section. There are some minor details that can increase the visibility for the general reader in this part: what was the mean CAL? What was the mean of gingival recessions? Furcation involvement? And according to what classification..... “ manual probe”- what type of periodontal probe was used in the study group?
Response 4: Thank you for your remark in this part. We have modified our material and methods. We amended 2.1. Study design and recruitment (page 3). It reads now as follows:
During the first visit, periodontitis patients underwent a full-mouth periodontal examination performed by periodontists or residents from the Department of Periodontology. The following measurements were carried out for 6 sites per tooth using a manual periodontal probe (Williams probe, PW6, HuFriedy, Chicago, IL, USA): probing pocket depth (PPD, distance from the gingival margin to the bottom of the pocket), gingival recessions (exposure of root surfaces due to apical migration of the gingival margin), clinical attachment loss (CAL, loss of the connective tissue attachment between the tooth and the supporting structures). Furthermore, tooth mobility (movement of the tooth horizontally and/or vertically) and furcation involvement (when supporting bone is lost around the branching point of a tooth with multiple roots) were measured.
Comment 5: Results are clearly presented, no modifications are required
Response 5: Thank you for reviewing this section!
Comment 6: There are some large paragraphs that require some referencing to support the statements. The authors should also add the strengths of the present research. Please also check the iThenticate report, since it’s quite high.
Response 6: Thank you for your remark. We have revised our paragraphs and added references where needed. Also we changed the structure of the discussion and added extra paragraphs, including the strengths of the present research in different sections of our Discussion (page 13). It reads now as follows:
The current study provides new insights in the prevalence of MetS and T2DM among dental attenders with LocPD, with GenPD and without periodontitis, utilizing the World Workshop 2017 classification criteria for periodontitis [14]. Over half of our study participants of ≥40 years, and even a higher proportion among those with periodontitis, are affected by MetS.
Furthermore, in comparison with the Dutch T2DM prevalence rate of 5.0% among patients (40-59 years) that visit their general physician, the prevalence rates of T2DM in our study were higher for all study groups [48].
Also, the mean age at which T2DM is diagnosed in the Netherlands is 60.3 years, whereas in our study, the mean age of patients with an elevated HbA1c level who were unaware of T2DM was 56.7 years [48]. This indicates that patients might be screened earlier in a dental office than in a general practice. Moreover, the results of this study showed that approximately one fifth of the periodontitis patients have elevated HbA1c levels.
In addition to potential of screening for underlying systemic conditions, dental professionals can play a crucial role in reminding patients, who are already aware of their condition, to consult their doctor when abnormal blood values are detected.
A strength of this study is the use of four different definitions of MetS. Previous research has demonstrated that different definitions of MetS may result in substantially different MetS prevalences [57]. To be able to compare the MetS prevalence with other studies, it is necessary to use the same criteria to define MetS, however a golden standard definition for MetS is still lacking. Another strength is that we also looked at the difference in prevalence among the extent of periodontitis: LocPD and GenPD. The prevalence of MetS in GenPD patients was higher compared to the control group only (according to the revised NCEP-ATP III) and compared to the controls combined with LocPD patients (according to the revised NCEP-ATP III and its updated definition). We suggest that patients with a large inflammatory burden may be more prone to have higher values for clinical cardiometabolic risk factors and may suffer more often from MetS [32,58].
Comment 7: Reference style is not consistent, please revise. Check also the format of the year of publication: most are written in bold, some are not.
Response 7: Thank you for your comment. We have revised our references and changed the bold format into a normal format.
|
